Methods

# Pathway-driven analysis of synthetic lethal interactions in cancer using perturbation screens

Mina Karimpour[1], Mehdi Totonchi[2,3], Mehrdad Behmanesh[1], Hesam Montazeri[4]

Synthetic lethality offers a promising approach for developing effective therapeutic interventions in cancer when direct targeting of driver genes is impractical. In this study, we comprehensively analyzed large-scale CRISPR, shRNA, and PRISM screens to identify potential synthetic lethal (SL) interactions in pan-cancer and 12 individual cancer types, using a new computational framework that leverages the biological function and signaling pathway information of key driver genes to mitigate the confounding effects of background genetic alterations in different cancer cell lines. This approach has successfully identified several putative SL interactions, including *KRAS-MAP3K2* and *APC-TCF7L2* in pan cancer, and *CCND1-METTL1*, *TP53-FRS3*, *SMO-MDM2*, and *CCNE1-MTOR* in liver, blood, skin, and gastric cancers, respectively. In addition, we proposed several FDA-approved cancer-targeted drugs for various cancer types through PRISM drug screens, such as cabazitaxel for *VHL*-mutated kidney cancer and alectinib for lung cancer with *NRAS* or *KRAS* mutations. Leveraging pathway information can enhance the concordance of shRNA and CRISPR screens and provide clinically relevant findings such as the potential efficacy of dasatinib, an inhibitor of *SRC*, for colorectal cancer patients with mutations in the WNT signaling pathway. These analyses revealed that taking signaling pathway information into account results in the identification of more promising SL interactions.

## Introduction

Cancer cells are characterized by unrestrained proliferation and dysregulated growth, which lead to the formation of malignant neoplasms (Hanahan & Weinberg, 2023). The development and progression of cancer have been linked to the dysregulation of multiple signaling pathways, including MAPK/ERK, Wnt/$\beta$-catenin, PI3K/AKT/mTOR, and NF-kB, which are among crucial pathways in cancer biology. The activation of these pathways is often driven by genetic alterations in the driver genes of the related pathways, leading to the emergence of different cancer phenotypes (Sanchez-Vega et al, 2018). Selectively targeting driver genes is a key objective in the development of cancer therapeutics, but it may not always be feasible, particularly for loss-of-function mutations in tumor suppressor genes (TSGs), which require restoration of their activity or the initiation of the signaling pathways they regulate (Liu et al, 2015). Targeting synthetic lethal (SL) gene partners that induce cancer cell death in the presence of specific driver genes is a promising alternative approach.

Synthetic lethality refers to a phenomenon in which the simultaneous presence of two genetic perturbations results in cell death, although each perturbation alone is insufficient to cause this outcome (Hartwell et al, 1997; Brough et al, 2011). In particular, SL interactions in cancer involve a mutated driver gene that contributes to cancer's growth and survival and a partner gene whose inhibition leads to cell death. Synthetic lethality offers a promising therapeutic approach, as inhibition of SL partner genes can selectively suppress proliferation in cancer cells with specific driver gene mutations, whereas sparing normal cells without these alterations (Ashworth & Lord, 2018). One established example of synthetic lethality in clinical oncology is the use of PARP inhibitors for the treatment of breast and ovarian cancers harboring *BRCA* mutations. The loss of function of the BRCA protein is compensated by the PARP protein, making PARP inhibitors a promising therapeutic approach that induces cell death through the principle of synthetic lethality (Miki et al, 1994; Wooster et al, 1995; Powell & Kachnic, 2003).

Several computational methods have been developed to identify SL interactions in human cancers using various high-throughput data sources. Some well-known approaches rely on data from SL interactions available in a well-established species (Jacunski et al, 2015), human metabolomics data (Folger et al, 2011) or human primary tumor data (Sinha et al, 2017) to construct network models and algorithms for identifying potential SL interactions. Achilles and DRIVE projects have employed genome-scale CRISPR knockout and shRNA knockdown screens, respectively, to explore the genetic vulnerabilities across hundreds of cancer cell lines (McDonald et al,

[1]Department of Genetics, Faculty of Biological Sciences, Tarbiat Modares University, Tehran, Iran   [2]Department of Genetics, Reproductive Biomedicine Research Center, Royan Institute for Reproductive Biomedicine, ACECR, Tehran, Iran   [3]Basic and Molecular Epidemiology of Gastrointestinal Disorders Research Center, Research Institute for Gastroenterology and Liver Diseases, Shahid Beheshti University of Medical Sciences, Tehran, Iran   [4]Department of Bioinformatics, Institute of Biochemistry and Biophysics, University of Tehran, Tehran, Iran

Correspondence: hesam.montazeri@ut.ac.ir; Behmanesh@modares.ac.ir

2017; Tsherniak et al, 2017). However, using data generated from such experiments to identify potential SL interactions poses several challenges, such as variable efficacies of targeting the same gene using different sgRNAs or shRNAs. Therefore, several computational techniques have been developed to aggregate multiple reagent scores into a gene-level dependency score that reflects the phenotype arising from the perturbation, whereas minimizing the technical biases. For CRISPR screens, CERES and CRISPRcleanR are two algorithms designed to tackle the problem of DNA cutting toxicity by accounting for copy number alterations that may impact the measured effect of sgRNAs (Meyers et al, 2017; Iorio et al, 2018). Recently, Chronos has been introduced as an explicit model of cell population dynamics to address multiple limitations of analyzing the CRISPR-Cas9 data (Dempster et al, 2021). Similarly for RNAi screens, the ATARiS algorithm summarizes the results of RNAi reagents targeting a gene by finding a subset of reagents with concordant phenotypic patterns across cell lines (Shao et al, 2013). DEMETER2 is another computational method that uses a hierarchical Bayesian inference for enhanced estimation of gene dependency scores. This method incorporates several confounding factors of RNAi screens and cell line screen-quality parameters into the model (McFarland et al, 2018).

Various computational tools have been introduced to identify SL interactions using large-scale shRNA and CRISPR screens. The DAISY framework leverages shRNA screening data, along with tumor copy number and expression data from human cancer cell lines, to predict potential SL interactions (Jerby-Arnon et al, 2014). ISLE is another computational approach that employs experimentally identified SL pairs and incorporates tumor molecular profiles, patient clinical data, and gene phylogeny to recognize clinically relevant SL interactions (Lee et al, 2018). More recently, Srivatsa et al (2022) proposed the SLIdR statistical algorithm based on Irwin Hall distributions, which employs shRNA perturbation screens to predict potential pan-cancer and cancer-specific SL pairs, even with small sample sizes (Srivatsa et al, 2022).

Several studies in the field of cancer diagnosis and treatment frequently investigate genetic alterations through the examination of signaling pathways (Peng et al, 2019; Alzahrani et al, 2023). This approach is preferred because many genetic alterations affect various components within a pathway, thereby exerting an influence on signaling pathway activity. Here, we present a computational framework that integrates signaling pathway information to identify potential SL interactions in both pan-cancer and individual cancer types. To this end, we stratified cancer cell lines into mutant and wild-type (WT) groups based on the driver gene status, followed by incorporating KEGG pathway information to filter out WT cell lines with mutations in other TSGs or oncogenes (ONGs) within the same pathway of the driver gene. We subsequently used the SLIdR statistical framework to compare the perturbation effects of each gene between mutant and pathway WT (pWT) cell lines. Our approach identified putative SL interactions in both pan-cancer and individual cancer types, and also suggested more clinically relevant SL interactions compared with not using pathway information. We also applied this framework to PRISM drug screens and discovered several FDA-approved cancer-targeted therapy drugs for different cancer types with specific genetic alterations. Furthermore, we performed a detailed assessment of potential SL

interactions in colorectal cancer (CRC) and demonstrated that *SRC*, along with its inhibitor dasatinib, could serve as a potential SL partner for mutations in the WNT signaling pathway.

# Results

### General workflow

The step-by-step framework for identifying potential SL interactions is illustrated in Fig 1. First, the necessary data, including CRISPR knockout and shRNA screens, PRISM drug screening data and pathway network elements for various cancer types, were obtained from relevant databases. In the next step, the cancer cell lines were classified into either mutant or WT groups based on the mutation and copy number status of the considered driver gene (see the Materials and Methods section). In our grouping approach, we examined tumor suppressor driver genes for the presence of damaging mutations or copy number deep deletions, whereas we assessed ONG driver genes for missense mutations and copy number amplifications. Furthermore, we eliminated cell lines with damaging or missense mutations in the pathway of the driver gene from the WT group and obtained the pWT cell lines to maximize the likelihood that the pathway of the driver gene is functionally normal in considered WT cell lines. To identify potential SL interactions, we first calculated normalized ranks of the viabilities of all perturbed genes for each cell line. Then, we applied the SLIdR statistical framework to identify genes with low viabilities in mutated cell lines. This framework retains the SL pairs where the viabilities of pWT cell lines are similar to those of healthy cells. In the final step, we analyzed the drug screening data from the PRISM project screen to determine whether inhibiting the SL partner with its targeted inhibitor could selectively reduce the viabilities of mutated cell lines compared with those of pWT cell lines.

### Characteristics of different datasets used for identification of SL interactions

We considered three main datasets to discover SL interactions in our computational pipeline. CRISPR and shRNA screens consist of viability scores for 1,054 and 398 cell lines from different cancer types, respectively. Fig 2A illustrates the number of cell lines available for different cancer lineages in CRISPR and shRNA screens. Pathways information from the KEGG database was considered as specific network elements associated with either pan-cancer or different cancer types (Table S1). Pathways of ERK, PI3K, WNT, Hedgehog, NOTCH, TGFB, JAK-STAT, calcium, HIF-1, cell cycle, apoptosis, KEAP1, nuclear receptor, telomerase activity were among those considered for the pan-cancer analysis (hsa05200; Fig 2B). Detailed information about these pathways for each cancer type and their associated genes can be found in Table S2. We refer to key ONGs and TSGs as driver genes throughout the article. Driver genes were selected from genes in KEGG cancer-specific pathway network elements, which are also categorized as tumor suppressor or ONG by TSGene or ONGene databases (Table S3).

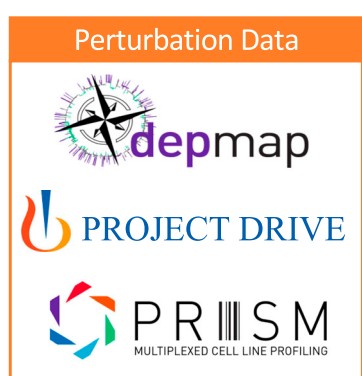
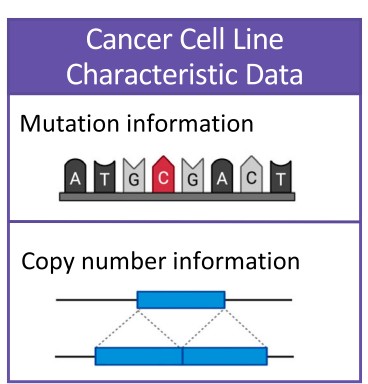
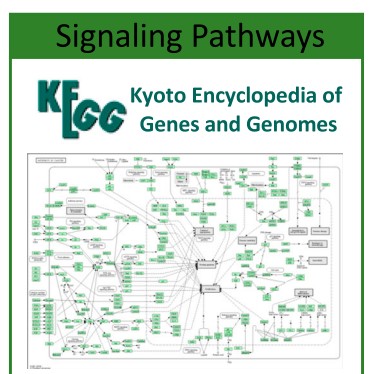
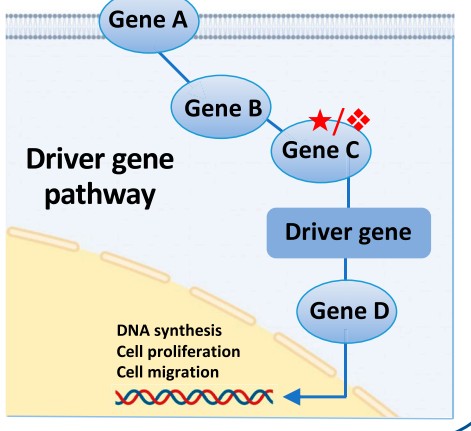
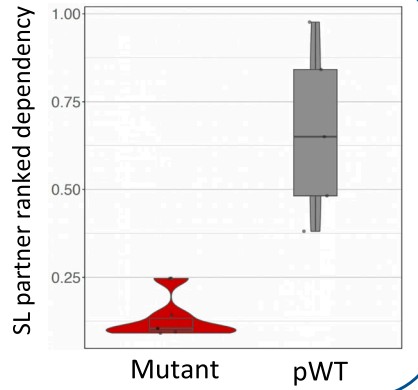

**Figure 1. General workflow for identifying SL interactions.**
We first obtained CRISPR screening data from DepMap, shRNA screening data from project DRIVE, drug screening data from the PRISM project, mutational and copy number characteristics of cancer cell lines from CCLE, and signaling pathway information of different cancer types from KEGG. For each driver gene, cancer cell lines are grouped based on the driver gene's mutational and copy number status into mutant and WT groups. Cell lines with additional mutations in the pathway of the driver gene are excluded from the WT group, resulting in the pWT group. We then employed the SLIdR statistical framework to identify potential SL partners for each driver gene. Finally, the $t$ test is used to detect SL partners whose perturbation could significantly differentiate the viabilities in the mutant and pWT groups.

 illustrates an oncoplot of the main driver genes grouped and ordered by the frequently mutated biological processes in 307 cell lines available in both screens, out of which, 275 cell lines (89.58%) harbor somatic alterations in the analyzed driver genes. The most frequent mutations were observed in *TP53* (57%), *KRAS* (24%), *PIK3CA* (19%), and *APC* (17%) resulting in the misregulation of

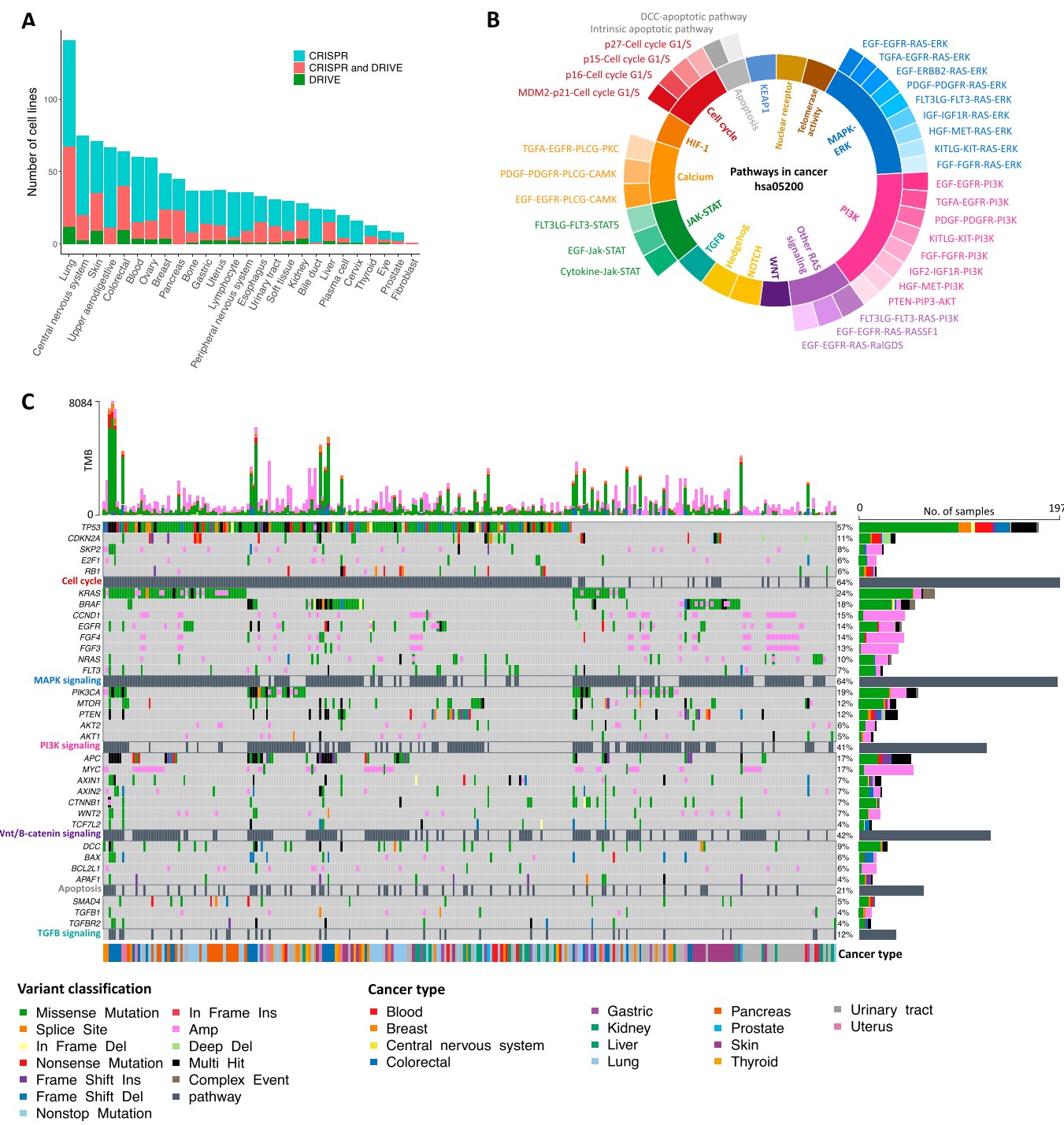

**Figure 2.   The characterization of cancer cell lines for SL identification in perturbation screens.**
**(A)** The number of cell lines exclusively in CRISPR (blue), exclusively in shRNA (green), and in both screens (pink) across 26 cancer lineages. **(B)** Specific pathways and pathway network elements related to pan-cancer (hsa05200) from the KEGG database. **(C)** An oncoplot illustrating the most frequently mutated genes and associated pathways, arranged in descending order of mutation frequency. It depicts the presence of SNPs, INDEL mutations, and copy number variations in common cancer cell lines of the CRISPR and shRNA screens. The top bar plot shows the frequency of different variant classifications across cell lines, whereas the right bar plot indicates the frequencies of various variant classifications in each gene. In addition, the cell lines are annotated according to their respective cancer types.

associated pathways. In addition, cell cycle, MAPK, WNT, and PI3K signaling pathways were frequently mutated with mutation percentages of 64%, 64%, 42%, and 41%, respectively. Furthermore, the

oncoplot highlights that TSGs including *TP53* and *APC* undergo different types of damaging mutations, such as nonsense, frame-shift insertion, and deletions, whereas ONGs such as *KRAS*, *BRAF*,

and *PIK3CA* have mainly missense and amplification alterations. Annotation of cancer cell lines by lineage reveals that alterations in tumor suppressor and ONGs may be enriched in specific cancer types because of their unique functions and the varying effects of mutations on these functions. The oncoplots for all cell lines used in the CRISPR and shRNA screens are also displayed in Fig S1A and B, respectively.

## Uncovering novel SL interactions in pan-cancer analysis from CRISPR and shRNA screens

In the pan-cancer analysis, we examined 51 TSGs and 76 ONGs across 40 pathway networks (Table S2). Figs 3A and S2A illustrate the number of mutant cell lines and WT cell lines with respect to the driver gene and the involved pathway in the CRISPR and shRNA screens, respectively. In addition, Fig S3A and B present a stacked barplot illustrating the distribution of the number of cell lines with a specific number of mutations (ranging from one to seven) within each signaling pathway for both CRISPR and shRNA projects. In the analysis of the CRISPR screen, we applied the proposed SL identification workflow on 97 driver genes and 17,225 perturbed genes in 993 cancer cell lines. A total of 194 SL pairs were found to be statistically significant, of which, 94 pairs appeared after leveraging pathway information. Similarly, the ranked viabilities of 522 out of 827 shRNA SL partners were significantly lower in their corresponding mutant groups versus pWT groups, specifically. Detailed information on all CRISPR and shRNA candidate SL interactions are listed in Tables S4 and S5, respectively. Top 120 SL pairs that were obtained after leveraging pathway information are shown in Sankey diagrams for CRISPR (Fig 3B and Table S4) and shRNA pan-cancer pairs (Fig S2B and Table S5). Our study identified several previously validated pairs, such as *KRAS-ID1* (Román et al, 2019), *KRAS-ITGA3* (Vizeacoumar et al, 2013), *BRAF-MAPK1* (Ko et al, 2020; Hicks et al, 2021), and *EGFR-SMAD2* (Chang et al, 2016), and newly identified putative SL pairs, such as *TP53-TP53BP1*, *APC-TCF7L2*, and *KRAS-MAP3K2*, which, to the best of our knowledge, have not been previously reported. The *TP53-TP53BP1* pair emerged as the top SL pair in the CRISPR pan-cancer analysis (Table S4 and Fig S4A and B). Targeting *TP53BP1*, which is regulated by TP53, is involved in DNA repair and enhances *TP53*-mediated transcriptional activation (Chen et al, 2007), may offer a potential therapeutic approach for *TP53*-mutated cancers. The analysis of the shRNA screen suggests that inhibiting *TCF7L2* substantially reduces dependency scores in *APC*-mutated cell lines (Fig S4C and D). Loss-of-function mutations in *APC*, a TSG, activate WNT signaling transduction and its target genes via TCF/LEF transcription factors (Liu et al, 2022). Therefore, targeting *TCF7L2* as a specific transcription factor in the WNT signaling pathway could potentially block the WNT pathway and reduce proliferation in *APC*-mutated cell lines.

Our method also identified *KRAS-MAP3K2* as a potential pan-cancer synthetic lethality pair by comparing *KRAS* pWT cell lines with *KRAS* mutant cell lines in the CRISPR screen. Because RAS ONGs are altered in about 10–30% of human cancers and directly targeting them is extremely challenging, discovering SL partners for RAS genes could offer a promising approach to cancer treatment (Prior et al, 2020). The waterfall plot of *MAP3K2* dependency showed enrichment of negative dependency scores in *KRAS* mutated cell

lines, specifically compared with *KRAS* pWT cell lines (Fig 3C). The *t* test also confirmed that dependency scores of *MAP3K2* are significantly lower in cell lines with *KRAS* mutation compared with *KRAS* pWT cells (Fig 3D). The PRISM data provide further evidence of SL interactions between *KRAS* and *MAP3K2*. Specifically, the *MAP3K2* inhibitor (PD-184352) demonstrated significantly higher sensitivity in *KRAS*-mutated cell lines compared with *KRAS* pWT cell lines (Fig 3E). In conclusion, targeting *MAP3K2* in *KRAS*-mutated cancer cell lines could affect cellular growth and viability, making it a potential SL pair for further in vitro and in vivo confirmation.

## Discovering cancer-specific SL interactions using pathway-informed analysis from CRISPR and shRNA screens

In our analysis, we examined 14 cancer types namely blood, breast, central nervous system, colorectal, gastric, kidney, liver, lung, pancreas, prostate, skin, thyroid, urinary tract, and uterus for which pathway information is available in KEGG database (Table S1). We could not classify the prostate and thyroid cancer cell lines based on driver gene statuses because of insufficient number of cell lines (less than three) in the Mut and pWT groups. The obtained SL pairs for individual cancer types from CRISPR and shRNA screens are presented in Tables S4 and S5, respectively. To assess the effects of leveraging pathway information in individual cancer types, we calculated the Spearman's correlation between CRISPR and shRNA-inferred *P*-values for the driver gene mutant, sMut, WT, and pWT groups in both pan-cancer and different cancer types (Table S6 and Fig 4A). Fig S5A and B display scatter plots for mutant and pWT groups, respectively, comparing *P*-values obtained from CRISPR and shRNA screens in different cancer types. These findings suggest that highlighting the pathway of the driver gene in the WT group rather than solely the driver gene could overall improve the correlation between the inferred *P*-values of two independent screens (Fig 4A). We additionally examined experimentally identified SL pairs from different in vitro screens (Han et al, 2017; Shen et al, 2017; Horlbeck et al, 2018; Najm et al, 2018; Zhao et al, 2018, 2019; Norman et al, 2019; Dede et al, 2020; DeWeirdt et al, 2020; Thompson et al, 2021), including combinatorial CRISPR, shRNA, and drug screens (see the Materials and Methods section), to determine overlap with our obtained SL pairs, with and without incorporating pathway information. In both CRISPR and shRNA screens, incorporating pathway information in the WT group led to a higher number of overlapping SL pairs (Fig S6A and Table S7). Interestingly, excluding mutated cell lines with mutations in other genes of the considered pathway (Mut-WT vs sMut-WT) led to a reduction in the number of experimentally identified interactions (Fig S6B).

Cancer-specific SL pairs are visualized as Sankey plots in Figs S7A–C, S8A–C, S9A–C, and S10A–C for individual cancer types. The plots reveal that the pathways associated with several SL pairs are shared across multiple cancer types, such as ERK, PI3K, and cell cycle pathways. Conversely, specific pathways are linked to certain SL pairs in particular cancer types, such as the HIF-1 signaling pathway in kidney cancer and the WNT signaling pathway predominantly in gastric and liver cancers. We discovered several SL pairs that have been previously reported in the literature, such as *BRAF-MAP2K1* in CRC (Salama et al, 2020; Klute et al, 2022), *KRAS-GRAP2* in lung cancer (Luo et al, 2009), and *MYC-UBE2O* in breast

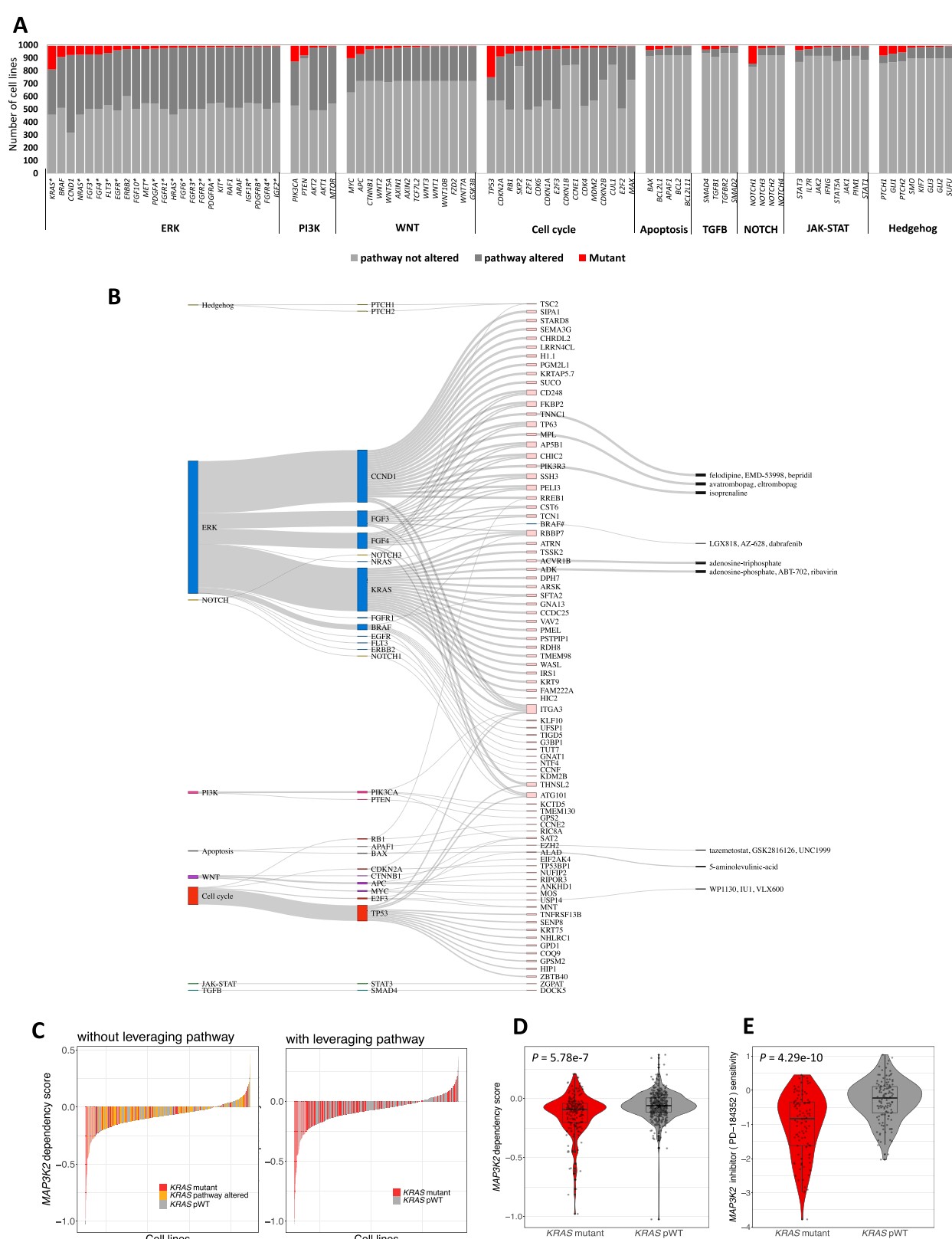

**Figure 3. The pan-cancer analysis of SL interactions using the CRISPR screen.**

**(A)** Stacked bar plot showing the frequency of top driver genes, grouped by relevant pathways, across cancer cell lines in the CRISPR screen. For each driver gene, the number of mutated cell lines (red), WT cell lines when the associated pathway is altered (dark gray), and WT cell lines when the associated pathway is not altered (light gray) are displayed. Note: genes marked with (*) in the ERK pathway are also involved in the PI3K pathway. **(B)** The specific SL interactions obtained by incorporating the

cancer (Kessler et al, 2012) and novel putative SL pairs, such as *TP53-FRS3* in blood cancer and *CCND1-METTL1* in liver cancer from CRISPR analysis. The waterfall and box plots demonstrate that inhibition of *FRS3* in blood cancer and *METTL1* in liver cancer leads to a significant decrease in the viability of *TP53* and *CCND1*-mutated cell lines, respectively, compared with pWT cell lines (Fig 4B and C). In addition, *CCNE1-MTOR* (gastric cancer) and *SMO-MDM2* (skin cancer) are found to be putative SL interactions according to the shRNA data.

### Identifying targetable SL interactions with FDA-approved cancer-targeted drugs

Incorporating pathway information allowed us to identify more drug-targetable SL partners for key driver genes, compared with the absence of pathway information (Fig S6C). We have identified several cancer-specific SL pairs whose interacting partner inhibitors are FDA-approved for the corresponding cancer types (Table 1). Table S8 provides information on FDA-approved cancer-targeted therapy drugs for specific types of cancer (National Cancer Institute, 2023). Our approach has identified FDA-approved cancer-targeted drugs that could potentially be used for treating other types of cancer (Table 1). For example, tamoxifen, an FDA-approved inhibitor of the protein kinase C genes *PRKCB* and *PRKCE*, currently used for breast cancer treatment, could potentially be used to target *EGFR or KRAS*-mutated lung cancers and *NRAS*-mutated skin cancer. We further analyzed the PRISM dataset to examine whether the inhibition of SL partners with their associated FDA-approved inhibitors could enhance the sensitivity in mutated cell lines compared with pWT cell lines in the suggested cancer types. Our analysis revealed that regorafenib and tivozanib significantly improved the sensitivity in *NRAS*-mutated skin and lung cancers, respectively (Fig 4D and E). These findings offer additional insights on how FDA-approved cancer-targeted drugs can be repurposed beyond their current clinical use for specific cancer types.

### PRISM dataset analysis

We applied the proposed framework on the drug perturbation PRISM screen to identify potential SL drugs for key driver genes in both pan-cancer and various cancer types. We aimed to identify drugs that significantly increase the sensitivity of mutated cell lines compared with pWT cell lines (mut $q$-value < 0.05, pWT $q$-value > 0.05) and also differentiate between mutant and pWT cell lines ($t$ test $P$-value < 0.05). We found multiple SL interactions between specific mutations and drugs that target genes other than the mutant driver gene (Table S9). In addition, our investigation identified several targeted therapy drugs that displayed efficacy in their target mutated cell lines, including taselisib and alpelisib

(PIK3CA inhibitor) in *PIK3CA*-mutated CRC, and canertinib and dacomitinib (EGFR inhibitors) in *EGFR*-mutated lung cancer.

We then focused on FDA-approved cancer-targeted therapy drugs, for which the effects on various cancer cell lines are available in the PRISM screen (Table S8). By considering pathway information, we were able to detect 10 out of 15 FDA-approved targeted drugs using the SLIdR framework for FDA-recommended cancer types with specific genetic alterations (mut $P$-value < 0.05, pWT $P$-value > 0.05). These included afatinib, erlotinib, and osimertinib for lung cancer, cobimetinib and vemurafenib for skin cancer, dabrafenib for thyroid cancer, alpelisib for breast cancer, and trametinib and dabrafenib for pan-cancer (Table S10). Two additional targeted drugs, dabrafenib and trametinib, were also detected for skin cancer with $P$-values close to the threshold for the pWT group. In contrast, only three cancer-targeted drugs were detected when pathway information was not considered (Table S10).

We extracted the statistically significant SL interactions from the PRISM screen analysis for FDA-approved cancer-targeted drugs (Table 2). These drugs can potentially act as SL partners for specific genetic alterations in suggested cancer types.

### *SRC* as a potential SL partner for key driver genes of the WNT pathway in CRC

In this section, we conducted an analysis of SL identification in CRC, which is the third most commonly diagnosed malignancy worldwide and the second leading cause of cancer-related deaths (Xi & Xu, 2021). Somatic genetic variations in *APC*, *KRAS*, *BRAF*, and *PIK3CA* genes are the most frequent in CRC, leading to the activation of key WNT–β-catenin and MAPK signal transduction pathways, which subsequently promote cell proliferation and cancer progression (Fig 5A). In total, 185 and 253 SL interactions were found in CRC using the CRISPR and shRNA screens, respectively (Tables S4 and S5). For top mutated driver genes in CRC, Fig 2B illustrates heatmaps of top SL pairs in CRISPR dataset based on $P$-values for each cell line group (Mut < 0.01 and pWT > 0.05). The heatmaps indicate that $P$-values are lower in the pWT group as compared with the WT group. Interestingly, excluding cell lines with mutations in genes other than the driver gene from the mutant group (sMut group; as defined in the Materials and Methods section) leads to higher $P$-values compared with the original mutant cell lines (Fig 5B). Similar heatmaps for CRISPR and shRNA SL pairs for key driver genes in CRC separated by their related signaling pathways are illustrated in Fig S11A and B.

Our computational workflow predicted the *SRC* gene as an SL partner for *TCF7L2* and *CTNNB1* genes, two key driver genes involved in the WNT signaling pathway, by analyzing the shRNA screen (Mut $q$-value < 0.01, pWT $q$-value > 0.1; Fig 5C). *SRC*, a non-receptor

pWT cell lines into the analysis pipeline. This Sankey plot illustrates each signaling pathway containing driver genes, with each driver gene connected to its corresponding SL partners. Each color represents a specific pathway according to Fig 2B. Furthermore, the druggable SL partners are connected to their inhibitors, as provided by the PRISM data. **(C)** The waterfall plots based on the dependency scores of *MAP3K2* knockout experiments. The left plot includes all cell lines and highlights *KRAS* mutant cell lines (red), *KRAS* pWT cell lines (gray), and cell lines with a WT *KRAS* but altered pathway (orange). The right plot is similar but excludes *KRAS* pathway mutant cell lines from the plot. **(D)** A box plot comparing the dependency scores of *MAP3K2* perturbation between the *KRAS* mutant and *KRAS* pWT cell lines in the CRISPR screen. **(E)** A box plot showing the sensitivity of *KRAS* mutant and *KRAS* pWT cell lines to PD-184352, the inhibitor of *MAP3K2*, in the PRISM screen.

**Figure 4. Results of pathway-informed SL identification workflow for different types of cancer.**
**(A)** A radar chart depicting the Spearman correlations, denoted by $r_s$, between the total sets of CRISPR and shRNA-derived SLIdR *P*-values for WT and pWT groups in specific cancer types and pan-cancer. **(B)** The waterfall plots show the CRISPR dependency scores of *FRS3* in blood cancer cell lines, colored according to the *TP53* status. The right box plot illustrates the comparison of *FRS3* perturbation dependency scores between *TP53* mutant and pWT cell lines in blood cancer. **(C)** The waterfall plots show the CRISPR dependency scores of *METTL1* in liver cancer cell lines, colored based on the *CCND1* status. The right box plot illustrates the comparison of *METTL1* perturbation dependency scores between *CCND1* mutant and pWT cell lines in liver cancer. **(D)** The left box plot shows a comparison between the CRISPR dependency scores of *RAF1* knockout in *NRAS*-mutated and *NRAS* pWT skin cancer cell lines. The right box plot displays the results of inhibiting *RAF1* with its inhibitor regorafenib in *NRAS*-mutated and pWT skin cancer cell lines in the PRISM screen. **(E)** The left box plot illustrates a comparison between the shRNA dependency scores for *FLT4* perturbation in *NRAS*-mutated and pWT lung cancer cell lines. The right box plot shows the impact of tivozanib, an FLT4 inhibitor, on both *NRAS*-mutated and pWT lung cancer cell lines in the PRISM screen.

tyrosine kinase, is a known proto-ONG that can activate multiple receptors involved in various cellular processes such as cell differentiation, cell adhesion, cell cycle progression, apoptosis, and migration (Simatou et al, 2020). Perturbation of *SRC* in *APC*-mutated CRC cell lines, the other crucial driver gene in WNT pathway, also significantly decreases the viability scores (*q*-value = 0.001) in comparison with the pWT group (*q*-value = 0.78) (Fig 5C). We also investigated the PRISM data for dasatinib, a potential inhibitor of *SRC* approved by FDA for leukemia, in the aforementioned driver genes of WNT pathway for mutated and pWT cell lines. We observed significantly higher sensitivity to dasatinib in CRC cell lines with mutations in *TCF7L2*, *CTNNB1* or *APC*, and *AXIN1*, which is another WNT pathway component, when compared with WNT pWT cell lines (Fig 5D). It should be noted that the *AXIN1-SRC* pair may potentially be classified as an SL pair. However, the mutant group fails to reach statistical significance after multiple testing corrections.

# Discussion

Cancer driver genes are characterized by genetic mutations that impair normal signaling pathways, thereby promoting neoplastic transformation. These genetic aberrations are central to the dysregulation of cellular signaling pathways, which govern critical cellular processes including proliferation, apoptosis, and metabolic regulation (Yip & Papa, 2021). The discovery of SL interactions that induce cell death in the presence of specific genetic alteration is a promising therapeutic approach to block uncontrolled cell proliferation. Several studies have used large-scale perturbation screens, such as CRISPR knockout and shRNA screens, to identify potential SL interactions, where the silencing of the partner gene results in decreased viability of cell lines with specific gene mutations compared with WT cell lines (De Kegel et al, 2021; Srivatsa et al, 2022). However, the heterogeneity of cancer cell lines makes it

**Table 1. List of potential SL interactions for targeted use of FDA-approved cancer-targeted drugs in various cancer types with specific genetic alterations other than those recommended by the FDA.**

| Driver gene | SL partner | Cancer type | FDA-approved targeted drug (recommended cancer type) |
|---|---|---|---|
| APC | KDR | colorectal | regorafenib (colorectal, liver); sorafenib, and cabozantinib (kidney, liver, thyroid); axitinib, tivozanib, and pazopanib (kidney); lenvatinib (uterus, kidney, liver, thyroid); sunitinib (kidney, pancreas); ponatinib (blood); vandetanib (thyroid); neratinib (breast) |
| | TEK | colorectal | regorafenib (colorectal, liver); ponatinib (blood); vandetanib (thyroid) |
| KRAS | MAP3K2 | colorectal | bosutinib (blood) |
| | SRC | pancreas | ponatinib, bosutinib, and dasatinib (blood); vandetanib (thyroid) |
| | PRKCB | lung | tamoxifen (breast) |
| CCND1 | CDK4 | breast | abemaciclib, ribociclib, and palbociclib (breast) |
| | ESR1 | breast | fulvestrant, toremifene, and tamoxifen (breast) |
| | FGFR4 | breast | Erdafitinib (urinary tract); ponatinib (blood) |
| NRAS | RAF1 | skin | dabrafenib (lung, skin, pan-cancer, thyroid); regorafenib**\*** (colorectal, liver); sorafenib (kidney, liver, thyroid); vemurafenib (skin) |
| | PRKCE | skin | tamoxifen (breast) |
| | FLT4 | lung | regorafenib (colorectal, liver); sorafenib (kidney, liver, thyroid); axitinib, tivozanib* and pazopanib (kidney); lenvatinib (uterus, kidney, liver, thyroid); sunitinib (kidney, pancreas); vandetanib (thyroid) |
| PIK3CA | SRC | colorectal | ponatinib, bosutinib, and dasatinib (blood); vandetanib**\*** (thyroid); |
| | PLK4 | colorectal | axitinib**\*** (kidney) |
| | MAPK11 | colorectal, breast | regorafenib (colorectal, liver); |
| | CDK4 | breast | abemaciclib, ribociclib, and palbociclib (breast) |
| | ESR1 | breast | fulvestrant**\***, toremifene, and tamoxifen (breast) |
| TP53 | PIK3CG | blood | alpelisib (breast); idelalisib (blood) |
| | CDK6 | blood | abemaciclib, ribociclib, and palbociclib (breast) |
| | ESR1 | lung | fulvestrant, toremifene, and tamoxifen (breast) |
| EGFR | MAP3K2 | colorectal | bosutinib (blood) |
| | PRKCB | lung | tamoxifen (breast) |
| BRAF | MAP2K1 | colorectal | trametinib (lung, skin, pan-cancer, thyroid); bosutinib (blood) |
| CCNE1 | MTOR | gastric | everolimus (central nervous system, breast, kidney, pancreas); temsirolimus (kidney) |
| MYC | ITK | lung | pazopanib (kidney) |
| PTEN | CDK2 | CNS | bosutinib (blood) |
| VHL | PIK3CD | kidney | alpelisib (breast), idelalisib (blood) |

According to the analysis of the PRISM dataset, drugs marked with an (*) significantly sensitized cell lines with specified driver gene mutations compared with pWT cell lines.

challenging to accurately find SL partners for a specific gene mutation in different cancer types because of other genetic alterations, which could result in finding spurious associations.

To mitigate this issue, our approach uses pathway information in addition to the mutational status of the driver gene to enhance the accuracy of identifying potential SL interactions. Our findings indicate that stratifying cancer cell lines by incorporating pathway information substantially increases the identification of potential SL interactions. This approach revealed multiple clinically relevant SL interactions which involve SL partners that can be targeted by specific drugs, some of which are currently approved by FDA for particular cancer types, but they may also be effective in targeting

**Table 2.  List of potential cancer-specific SL interactions from PRISM screen analysis with FDA-approved cancer-targeted drugs.**

| Cancer type | Driver gene | FDA-approved targeted drug | Target |
|---|---|---|---|
| Breast | PTEN | temsirolimus | MTOR |
| CNS | NRAS | temsirolimus | MTOR |
| | RB1 | crizotinib | ALK, MET |
| Colorectal | PIK3CA | alpelisib | PIK3CA, PIK3CB, PIK3CD, PIK3CG |
| | DDB2 | gefitinib | EGFR |
| | PIK3CA | vandetanib | EGFR, EPHA1-8, EPHA10, EPHB1-4, EPHB6, ERBB2-4, FLT1, FLT4, KDR, PTK6, RET, SRC, TEK, VEGFA |
| | DDB2 | | |
| | PIK3CA | neratinib | EGFR, ERBB2, KDR |
| Gastric | CTNNB1 | cobimetinib | MAP2K1, MAP2K2 |
| | CTNNB1 | brigatinib | ALK, EGFR |
| Kidney | VHL | cabazitaxel | TUBA4A, TUBB, TUBB1 |
| Liver | CCND1 | cobimetinib | MAP2K1, MAP2K2 |
| Lung | KRAS | alectinib | ALK, MET |
| | NRAS | | |
| | CDK6 | crizotinib | ALK, MET |
| | EGFR | dacomitinib | EGFR, ERBB2, ERBB4 |
| Pancreas | TP53 | venetoclax | BCL2 |
| Skin | TP53 | regorafenib | ABL1, BRAF, DDR2, EPHA2, FGFR1-2, FLT1, FLT4, FRK, KDR, KIT, MAPK11, NTRK1, PDGFRA, PDGFRB, RAF1, RET, TEK |
| Urinary tract | E2F3 | gilteritinib | FLT3 |
| | CCND1 | regorafenib | ABL1, BRAF, DDR2, EPHA2, FGFR1, FGFR2, FLT1, FLT4, FRK, KDR, KIT, MAPK11, NTRK1, PDGFRA, PDGFRB, RAF1, RET, TEK |
| | CDKN1A | tivozanib | FLT1, FLT4, KDR, KIT, PDGFRA, PDGFRB |
| | CCND1 | trametinib | MAP2K1, MAP2K2 |
| | CDKN2A | | |
| Uterus | CTNNB1 | abiraterone | CYP11B1, CYP17A1 |
| | MYC | imatinib | ABL1, CSF1R, DDR1, KIT, NTRK1, PDGFRA, PDGFRB, RET |
| | APC | talazoparib | PARP2 |
| | APC | vandetanib | EGFR, EPHA1-8, EPHA10, EPHB1-4, EPHB6, ERBB2-4, FLT1, FLT4, KDR, PTK6, RET, SRC, TEK, VEGFA |

Alpelisib, a PIK3CA inhibitor, is suggested as targeted therapy for PIK3CA-mutated colorectal cancer, thus, PIK3CA–alpelicib cannot be considered as an SL interaction by definition. We exclude drugs that exhibit synthetic lethality with more than two driver genes in a specific cancer type because of the possibility of mutations in these genes overlapping across the evaluated cell lines.

other cancer types with specific altered driver genes. This improvement is because of the fact that mutations in multiple genes within a signaling pathway can have similar impacts on pathway regulation, resulting in similar cancer phenotypes (Sanchez-Vega et al, 2018). Hence, cell lines with mutations in other components of a signaling pathway, despite lacking mutations in the driver gene itself, can have a a similar cancer phenotype as cell lines with a mutated driver gene. Excluding such cell lines from the WT group and focusing on those with an intact signaling pathway results in a more pronounced difference compared with not considering pathway information. The obtained results suggest that incorporating additional biological processes could better reflect the biological characteristics of cancer cells and potentially lead to the identification of promising SL interactions.

The use of CRISPR-Cas9 and RNAi is widespread for the examination of loss-of-function phenotypes in cell lines; however, these methods have limitations that need to be considered. RNAi triggers, such as siRNA or shRNA, can cause off-target effects via an unintended selection of the passenger strand as the target strand by AGO2 (Schwarz et al, 2003) or through the entry of the small RNA into the miRNA pathway (Jackson et al, 2003; Kaelin, 2012), potentially leading to the silencing of unintended transcripts. The CRISPR-Cas9 system depends on short sequence-specific recognition and has the potential for off-target effects (Zhang et al, 2015). In addition, compensation mechanisms can influence the interpretation of loss-of-function studies using the CRISPR-Cas9 system as other genes or pathways may compensate for the loss of function of the

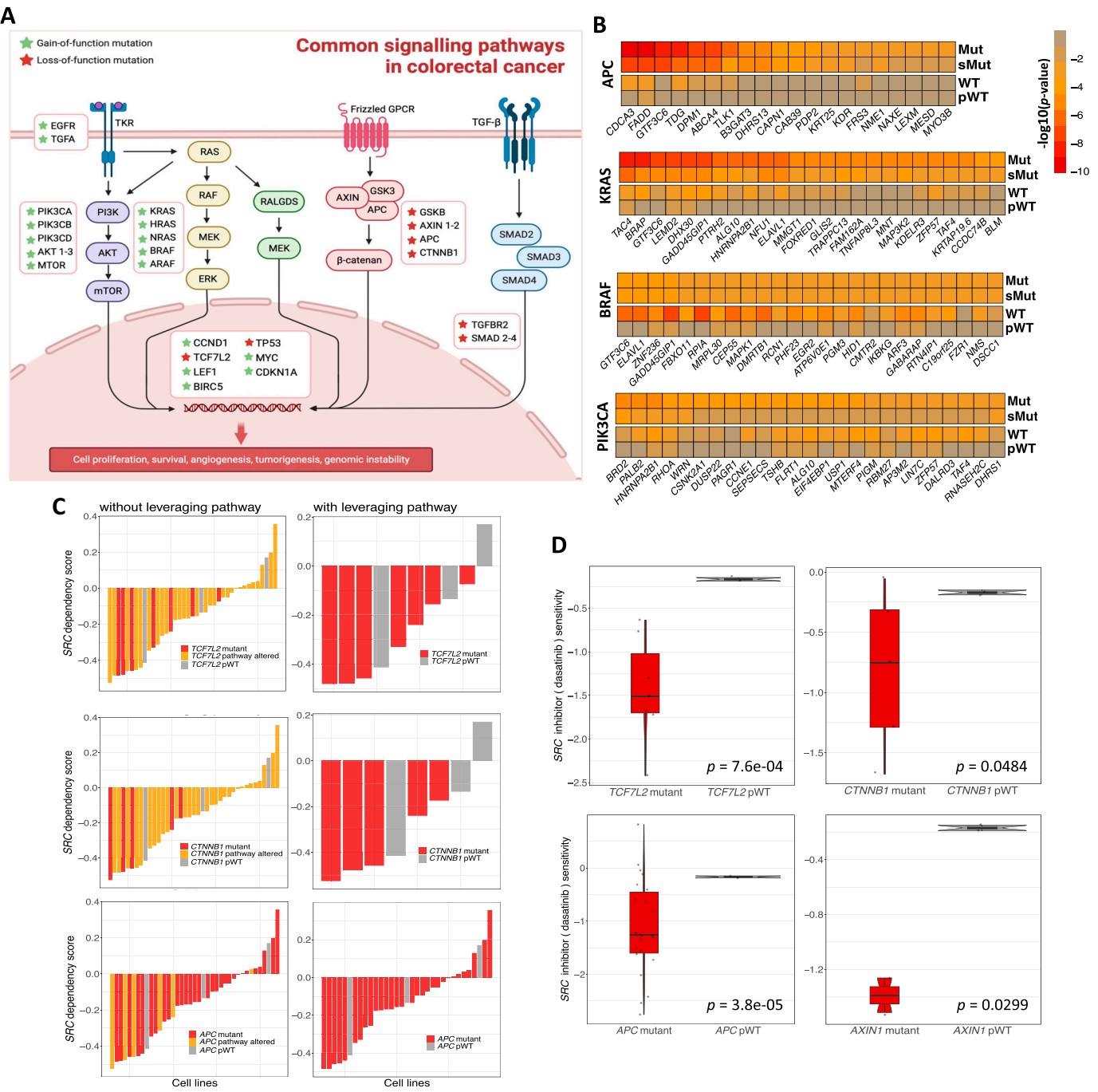

**Figure 5. Analysis of SL interactions in colorectal cancer.**
**(A)** A schematic image of common signaling pathways in CRC with associated TSGs and ONGs involved in each pathway. The figure is created with BioRender.com.
**(B)** Heatmaps for top SL interactions from CRISPR screens depicting the *P*-values of each SL partner for *APC*, *KRAS*, *BRAF*, and *PIK3CA* mutants, sMut, WT, pWT CRC cell lines.
**(C)** Waterfall plots for *TCF7L2-SRC*, *CTNNB1-SRC*, *APC-SRC* pairs in CRC cell lines with or without leveraging pathway information. Mutated cell lines are marked in red, whereas cell lines with WNT pWT are marked in gray. The orange segments represent the cell lines that are WT with respective driver gene, but the other elements of the WNT pathway are altered. **(D)** Boxplots illustrating the sensitivity of dasatinib, a potential inhibitor of SRC, in *TCF7L2*, *CTNNB1*, *APC*, and *AXIN1* mutant and pWT colorectal cancer cell lines.

targeted gene (Rossi et al, 2015). Emerging evidence has demonstrated that knockouts and knockdowns can result in diverse phenotypic variations across various model organisms (De Souza et al, 2006; Kok et al, 2015; Morgens et al, 2016). Our results are also concordant with previous findings, though we observed incorporating pathway information improves the correlation between CRISPR and shRNA screens, especially in cancer-specific contexts.

There are several limitations to our approach. Firstly, we solely relied on the KEGG database as a source of pathway information. This database does not have pathway information for certain cancer types such as bone, lymphocyte, esophagus, and soft tissue cancers, thereby limiting the scope of our analysis. Secondly, our approach of incorporating pathway information may result in a substantial reduction of the number of cell lines in the WT group, leading to the exclusion of several driver genes from statistical analysis. Thirdly, we employed gene-level scores from both CRISPR and shRNA perturbation screens to identify potential SL interactions. However, the limitations of computational algorithms used to calculate these scores could affect the accuracy of our results. Further research is required to refine existing reagent-level computational frameworks (Allen et al, 2019; Toghrayee & Montazeri, 2023 *Preprint*) such that they account for pathway information for finding cancer dependencies. Fourthly, our approach solely focused on individual components of signaling pathways to determine whether any alterations had occurred. However, taking into account the interactions among these components and how each impacts the function of others in the pathway may result in more precise and accurate findings. To address this issue, we used the diffusion algorithm used by Vandin et al (2011) on KEGG pathways to calculate *diffused* mutation status of the driver gene for each cell line using all mutations in the involved pathway. We subsequently used these diffused mutation statuses for determining WT cell lines. Despite our efforts, we did not observe any significant differences between our approach and the diffusion-based approach (obtained results are not shown here). Further research is required to explore alternative methods for integrating biological networks in finding SL interactions.

In conclusion, this study highlights the potential of incorporating pathway information in identifying SL pairs from large-scale perturbation screens in various cancer types. Our approach discovers several potentially clinically relevant SL interactions that may lead to novel mutation-specific personalized therapies in cancer.

# Materials and Methods

### Genomic perturbation data

We used the CRISPR gene-level viability scores from the DepMap project (21Q4), which contains the Achilles (Avana Cas9 library) and Sanger's project SCORE (KY Cas9 library) genome-scale CRISPR knockout screens (Tsherniak et al, 2017; Behan et al, 2019; Dwane et al, 2021). This dataset comprises Chronos gene effect scores of 17,387 genes in 1,054 cancer cell lines (Dempster et al, 2021). We also obtained DEMETER2 absolute gene dependency scores (McFarland et al, 2018) from the project DRIVE shRNA screens. In project DRIVE, 398 cancer cell lines were employed to target 7,975 genes by using an average of 20 pooled shRNAs per gene. The RSA viability scores were also used to specify essential genes which were removed if they had an RSA value of ≤−3 in more than half of the cell lines from the DRIVE dataset. The cell lines without information on gender and laboratory source, and those that had no mutation profile, were removed from both perturbation screens. To enable cross-

comparison of the viability scores across different cell lines, we ranked all the genes based on their viability scores upon perturbation in each cell line and then normalized the ranks to a range of 0 to 1 (Montazeri et al, 2021).

### Drug perturbation data

We collected drug sensitivity data from the Profiling Relative Inhibition Simultaneously in Mixtures, PRISM project (Corsello et al, 2020) through the DepMap data portal. We used the primary PRISM Repurposing 19Q4 screen, which includes data on viability screens for 4,686 compounds in 555 cancer cell lines. We excluded cell lines that lacked information on gender and laboratory source, and those with no mutation profile. We then ranked all drugs based on viability scores of cell lines, normalizing the ranks to a range of 0 to 1 (Montazeri et al, 2021).

The list of FDA-approved cancer-targeted therapy drugs, published by the National Cancer Institute (National Cancer Institute, 2023), were also used to discover potential FDA-approved targeted drugs for specific genetic alterations in different types of cancer.

### Characteristics of cancer cell lines

We used data from the Cancer Cell Line Encyclopedia (CCLE) project (Ghandi et al, 2019), available through the DepMap data portal (21Q4), to acquire the general information including the lineage, laboratory source, and gender, and somatic mutation profiles associated with cancer cell lines. To obtain copy number information, we downloaded the copy number segments file from DepMap and applied the GISTIC2 module (https://broadinstitute.github.io/gistic2/) to obtain five-level qualitative copy number values. The applied parameters were as follows: *genegistic 0, smallmem 1, maxseg 5000, savegene 1, saveseg 1, savedata 0, v 30, qvt 0.01, conf 0.99, broad 1, brlen 0.5, and rx 0*. The output file indicates copy number aberrations based on a prespecified $q$-value threshold, where values of "0," "1," and "2" denoting no, low-level, and high-level aberrations, respectively. Copy number losses are indicated by negative values, whereas gains are indicated by positive values. In this study, we considered values of "−2" to represent deep deletions and values of "2" to represent highly amplified genes.

### Pathways information

The information on signaling pathways in different cancer types was obtained from the Kyoto Encyclopedia of Genes and Genomes (KEGG) PATHWAY database (https://www.genome.jp/kegg/). By providing the KEGG pathway "hsa" identifiers as input (Table S1), we used the KEGGREST package in R to obtain relevant network element identifiers of KEGG pathways for individual cancer types. The obtained "network element" identifiers were then used as input to acquire the gene list of each signaling pathway. Detailed information on the network elements involved in specific cancer types, along with the corresponding gene list, is provided in Table S2. Note that the pathway network elements of acute myeloid leukemia (hsa05221) and chronic myeloid leukemia (hsa05220) have been grouped together as "blood cancer", and basal cell carcinoma (hsa05217) and melanoma (hsa05218) as "skin cancer" and small

cell lung cancer (hsa05222) and non-small cell lung cancer (hsa05223) as "lung cancer".

### Determining driver genes

To specify the key driver genes for pan-cancer and individual cancer types, we used the gene list obtained from pathway network elements. We then selected ONGs from the literature-based ONG database (Liu et al, 2017) and TSGs from the TSG database (Zhao et al, 2013). The list of selected TSGs and ONGs for individual cancer types with their associated pathways is provided in Table S3. This approach allows us to determine critical genes that undergo genetic alterations in specific cancer types, making it more valuable to identify their relevant SL partners.

### Grouping cancer cell lines

Cell lines were stratified into mutant and WT groups based on the mutational and copy number statuses of the driver gene obtained from CCLE project (21Q4) for cancer-specific and pan-cancer analyses, as explained below.

Firstly, for ONG drivers, cell lines with missense variations and amplifications were assigned to the mutant group. We exclusively considered missense mutations that were annotated as hotspots in both the TCGA and COSMIC datasets, as indicated in the somatic mutation information within the CCLE. Non-hotspot mutations were excluded from our analysis, as they are more likely to occur within a protein structure without significant clinical impact. For tumor suppressor driver genes, mutant cell lines were defined as those carrying any damaging mutations, such as "Nonsense Mutation," "Frame Shift Insertion," "Frame Shift Deletion," "Splice Site," "De novo Start Out of Frame," "Start Codon SNP," "Start Codon Deletion," "Start Codon Insertion," and copy number deep deletions. For driver genes that are classified as both ONGs and tumor suppressors, cell lines were placed in the mutant group if they exhibited either missense or deleterious mutations, and amplifications or deep deletions.

In the next step, we examined the WT group for other alterations in the pathway of the evaluated driver gene, assessing for missense variations and amplifications in ONGs, and damaging mutations in TSGs. WT cell lines having critical alterations in the driver gene's pathway were excluded from the analysis. We specifically examined all genes within any signaling pathway that included the given driver gene. After this process, three groups of cell lines were generated for pan-cancer and individual cancer types, including "driver gene mutant (Mut)," "driver gene WT" and "driver gene pWT" groups, which were employed in our subsequent analyses.

It is worth noting that the mutational status of each perturbed gene in the driver gene mutant group was evaluated for TCGA and COSMIC hotspot variations. Cell lines with mutations in both the driver gene and the perturbed gene were removed from the mutant group too.

### Statistical analysis

We applied the SLIdR framework (Srivatsa et al, 2022) to conduct statistical tests for finding SL interactions based on the viability scores of the aforementioned cell line groups. SLIdR, based on the Irwin-Hall distribution, employs a one-sided statistical test for mutant cell lines to evaluate whether the normalized ranks of viabilities resulting from knockout or knockdown of the SL candidate were significantly lower than would be expected by chance. This approach was relevant in the context of synthetic lethality, seeking to identify partners whose loss of function is only lethal in the presence of a specific driver gene mutation. Conversely, SLIdR uses a two-sided Irwin-Hall test for evaluated driver gene WT cell lines to ensure that the viability scores after perturbation of the SL candidate did not deviate significantly from the expected values, as WT cell lines are expected to behave similarly to healthy cells.

In the present study, we applied SLIdR to mutant and pWT cell lines of each possible driver gene in pan-cancer and individual cancer types followed by computing false discovery rate for multiple testing corrections. Ultimately, a perturbed gene was selected as an SL partner for a given driver gene only if the $q$-value of the SLIdR test was statistically significant in mutant cell lines ($q$-value < 0.01), but not in pWT cell lines ($q$-value > 0.1). The selected SL partners must demonstrate significantly lower mean normalized ranks in the mutant cell lines compared with the pWT cell lines ($t$ test $P$-value < 0.05) to qualify as a final SL interaction.

### CRISPR and shRNA inferred correlations

To assess the correlation between CRISPR and shRNA perturbation data after integrating pathway information, we conducted Spearman's correlation tests on the SLIdR $P$-values of all inferred common SL pairs from CRISPR and shRNA analyses for both pan-cancer and individual cancer types. This analysis was performed for mutant, single driver gene mutant (sMut), WT, and pWT groups.

### Comparison on experimentally identified SL interactions

To compare the predicted SL interactions with experimentally identified SL interactions, we used a dataset of 6,033 SL pairs reported by Lee et al, which were obtained from various in vitro screens (including shRNA, sgRNA, and drug screens) across different cancer types (Lee et al, 2018). In addition, we considered data collected from 10 different combinatorial CRISPR screens, comprising a total of 24,651 pairs (Srivatsa et al, 2022). We extracted the experimentally identified SL pairs that were analyzed in our pipeline, excluding those that were not inferred, for both pan-cancer and individual cancer types and determine the overlaps between predicted SL pairs (Mut or sMut $P$-value < 0.05 and WT or pWT $P$-value > 0.05) and experimentally validated SL interactions for Mut-WT, Mut-pWT, sMut-WT, and sMut-pWT groups. We considered experimentally identified SL pairs to be unordered (i.e., geneA-geneB and geneB-geneA are equivalent).

## Data Availability

The CRISPR gene effects from Projects Achilles and SCORE (21Q4), DEMETER2 dependency scores from Project DRIVE, and primary PRISM repurposing screen (19Q4) were obtained from the DepMap

data portal (https://depmap.org/portal/download/all/). In addition, molecular profiling data of the cancer cell lines from the Project CCLE, including the somatic mutations and the copy number segments (21Q4) were also provided from the DepMap data portal. Signaling pathway information for pan-cancer and individual cancer types was acquired from the KEGG database (https://www.genome.jp/kegg/). The lists of TSGs and ONGs are available in TSGene (https://bioinfo.uth.edu/TSGene/) and ONGene (http://ongene.bioinfo-minzhao.org) databases, respectively.

## Supplementary Information

## Acknowledgements

This project was supported by the Iran National Science Foundation and the Department of Research Affairs at Tarbiat Modares University.

### Author Contributions

M Karimpour: conceptualization, data curation, software, formal analysis, validation, visualization, methodology, and writing—original draft.
M Totonchi: investigation and writing—review and editing.
M Behmanesh: conceptualization, supervision, methodology, project administration, and writing—review and editing.
H Montazeri: conceptualization, supervision, methodology, and writing—original draft, review, and editing.

### Conflict of Interest Statement

The authors declare that they have no conflict of interest.

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
