## [Reviewer comments · Life Science Alliance]

Life Science Alliance

Pathway-driven Analysis of Synthetic Lethal Interactions in Cancer Using Perturbation Screens

Mina Karimpour, Mehdi Totonchi, Mehrdad Behmanesh, and Hesam Montazeri

DOI: <https://doi.org/10.26508/lsa.202302268>

Corresponding author(s): Hesam Montazeri, University of Tehran and Mehrdad Behmanesh, Tarbiat Modarres University

Review Timeline:

Submission Date:	2023-07-11
Editorial Decision:	2023-08-28
Revision Received:	2023-09-14
Editorial Decision:	2023-10-03
Revision Received:	2023-10-10
Accepted:	2023-10-10

Transaction Report:

August 28, 2023

Re: Life Science Alliance manuscript #LSA-2023-02268-T

Hesam Montazeri
University of Tehran, Department of Bioinformatics

Dear Dr. Montazeri,

Thank you for submitting your manuscript entitled "Identifying Synthetic Lethal Interactions in Cancer using CRISPR, shRNA, and Drug Perturbation Screens: A Signaling Pathway-Driven Approach" to Life Science Alliance. The manuscript was assessed by expert reviewers, whose comments are appended to this letter. We invite you to submit a revised manuscript addressing the Reviewer comments.

Thank you for this interesting contribution to Life Science Alliance. We are looking forward to receiving your revised manuscript.

Sincerely,

B. MANUSCRIPT ORGANIZATION AND FORMATTING:

Reviewer #1 (Comments to the Authors (Required)):

The authors introduce an innovative yet straightforward approach aimed at enhancing the identification of synthetic lethal gene pairs and pharmacogenomics interactions. They achieve this by analysing extensive public datasets derived from functional genetic and drug screens conducted on thousands of cancer cell lines. Notably, these datasets include comprehensive multi-omic characterisations that are openly accessible to the public.

In a nutshell, their method hinges on a systematic contrastive analysis of gene essentiality and drug responses between altered and wild-type cell lines. This contrast is established based on the status of a panel of cancer driver genes, encompassing both tumour suppressors (where loss-of-function mutations or copy number deletions are considered) and oncogenes (where gain-of-function mutations and copy number amplifications are considered). What sets this approach apart is the authors' strategy of excluding cell lines from the analysis if they do not harbour alterations in the specific cancer driver gene under scrutiny but rather in another gene that shares a biological pathway with the driver gene. They employ public functional annotations from the KEGG dataset for this purpose.

The manuscript is well crafted, featuring compelling visual aids. It successfully demonstrates the efficacy of the presented approach in not only confirming previously reported synthetic lethal gene pairs but also revealing novel ones.

In essence, this work is a valuable, simple yet ingenious contribution to the field, sure to pique the interest of life science alliance readers.

Before considering this manuscript for publication, a few minor points warrant attention:

- When discussing the use of CRISPR and shRNA knockout screens across cancer cell lines for identifying genetic vulnerabilities, it is advisable to include the following two citations: PMID: 33068406 and PMID: 30971826.

- Ensure consistency by referring to "CRISPRcleanR" with a lowercase "c" in "clean."

- Clarify how the "pathway of the driver gene" was determined. Is it merely the set of genes listed in at least one KEGG pathway alongside the driver gene of interest? Or does it involve a specific pathway shared with the oncogene? In the latter case, specify which pathway, as oncogenes can be part of various pathways.

- Address readability issues with Figure 3.

Reviewer #2 (Comments to the Authors (Required)):

Clearly written paper that uses pathway defined genes to increase the statistical power to detect Synthetic Lethal interactions from CRISPR/shRNA screen data from the DepMap consortium.

Comments

1. "Firstly, for oncogene drivers, cell lines with missense variations and amplifications were assigned to the mutant group. Missense variations in oncogenes were also checked for TCGA and COSMIC hotspots, as these may occur in a protein structure with no clinical effects."

Q. It's not clear how driver mutations were called - did the authors consider using the 'OmicsSomaticMutations.csv' file (<https://depmap.org/portal/download/all/?releasename=DepMap+Public+23Q2&filename=OmicsSomaticMutations.csv>) from the DepMap consortium and which annotates mutations with a 'Driver/hotspot' call? This would give a more up-to-date list of oncogene drivers.

2. The authors used KEGG as the source of pathways - were other Gene Ontologies tested (Hallmarks, Reactome etc) and if so, why was KEGG used?

3. "Our method also identified KRAS-MAP3K2 as a potential pan-cancer synthetic lethality pair by comparing ERK pWT cell lines with KRAS mutant cell lines in the CRISPR screen." - I don't understand this statement

4. The authors detect SL pairs without and with 'leveraging pathway' data e.g. Figure 3C - it's not clear to me how the pathway data is used, either from the main text or the methods - this needs to be explained more clearly - initially, the paper describes 9 specific pathways where genes are altered but then focuses entirely on specific genes

5. Figure 3a - for each of the 9 pathways, useful to indicate how many cell lines have a mutation in any gene in that pathway, in addition to the existing view which shows each specific gene

Typos

Abstract - 'MPA3K2' should be 'MAP3K2'

We appreciate the reviewers and the editor for their insightful feedback and the opportunity to revise the manuscript. We have carefully incorporated all comments and believe that the quality of the manuscript has improved following the revision. Furthermore, we have also made adjustments to adhere to the formatting guidelines. Please find below our detailed point-by-point responses.

Responses to comments by Reviewer #1

The authors introduce an innovative yet straightforward approach aimed at enhancing the identification of synthetic lethal gene pairs and pharmacogenomics interactions. They achieve this by analysing extensive public datasets derived from functional genetic and drug screens conducted on thousands of cancer cell lines. Notably, these datasets include comprehensive multi-omic characterisations that are openly accessible to the public.

In a nutshell, their method hinges on a systematic contrastive analysis of gene essentiality and drug responses between altered and wild-type cell lines. This contrast is established based on the status of a panel of cancer driver genes, encompassing both tumour suppressors (where loss-of-function mutations or copy number deletions are considered) and oncogenes (where gain-of-function mutations and copy number amplifications are considered). What sets this approach apart is the authors' strategy of excluding cell lines from the analysis if they do not harbour alterations in the specific cancer driver gene under scrutiny but rather in another gene that shares a biological pathway with the driver gene. They employ public functional annotations from the KEGG dataset for this purpose.

The manuscript is well crafted, featuring compelling visual aids. It successfully demonstrates the efficacy of the presented approach in not only confirming previously reported synthetic lethal gene pairs but also revealing novel ones.

In essence, this work is a valuable, simple yet ingenious contribution to the field, sure to pique the interest of life science alliance readers.

Before considering this manuscript for publication, a few minor points warrant attention:

- When discussing the use of CRISPR and shRNA knockout screens across cancer cell lines for identifying genetic vulnerabilities, it is advisable to include the following two citations: PMID: 33068406 and PMID: 30971826.

Thank you for the suggestion. We have now included the recommended citations (PMID: 33068406 and PMID: 30971826) into the "Genomic perturbation data" subsection of the Methods section.

- Ensure consistency by referring to "CRISPRcleanR" with a lowercase "c" in "clean."

We have made the necessary adjustments in the manuscript.

- Clarify how the "pathway of the driver gene" was determined. Is it merely the set of genes listed in at least one KEGG pathway alongside the driver gene of interest? Or does it involve a specific pathway shared with the oncogene? In the latter case, specify which pathway, as oncogenes can be part of various pathways.

We have identified all signaling pathways in the KEGG database associated with the driver gene and have incorporated all genes participating in these pathways into our analysis. We have elaborated on this procedure in the third paragraph of the "Grouping cancer cell lines" subsection of the Methods section, where we state

“Wild-type cell lines having critical alterations in the driver gene’s pathway were excluded from the analysis. We specifically examined all genes within any signaling pathway that included the given driver gene.”

- Address readability issues with Figure 3.

We have separately uploaded the high-quality versions of all figures.

Responses to comments by Reviewer #2

Clearly written paper that uses pathway defined genes to increase the statistical power to detect Synthetic Lethal interactions from CRISPR/shRNA screen data from the DepMap consortium.

1- *“Firstly, for oncogene drivers, cell lines with missense variations and amplifications were assigned to the mutant group. Missense variations in oncogenes were also checked for TCGA and COSMIC hotspots, as these may occur in a protein structure with no clinical effects.”*

Q. *It's not clear how driver mutations were called - did the authors consider using the 'OmicsSomaticMutations.csv' file (<https://depmap.org/portal/download/all/?releasename=DepMap+Public+23Q2&filename=OmicsSomaticMutations.csv>) from the DepMap consortium and which annotates mutations with a 'Driver/hotspot' call? This would give a more up-to-date list of oncogene drivers.*

We used the CCLE_mutations.csv from DepMap 21Q4, which includes annotations for hotspot alterations from TCGA and COSMIC databases. We clarified this in the manuscript “Grouping cancer cell lines” subsection of the Methods section where we write:

“We exclusively considered missense mutations that were annotated as hotspots in both the TCGA and COSMIC datasets, as indicated in the somatic mutation information within the CCLE. Non-hotspot mutations were excluded from our analysis, as they are more likely to occur within a protein structure without significant clinical impact.”

2- *The authors used KEGG as the source of pathways - were other Gene Ontologies tested (Hallmarks, Reactome etc) and if so, why was KEGG used?*

We chose KEGG due to its well-established quality and consistency, achieved through rigorous curation and regular updates. Moreover, KEGG pathway networks seamlessly integrated into our analytical framework in R. Nevertheless, we did not conduct a comprehensive comparison of different pathway sources in this manuscript. We acknowledge this limitation in the Discussion section, as we state,

“There are several limitations to our approach. Firstly, we solely relied on the KEGG database as a source of pathways information. This database does not have pathways information for certain cancer types such as bone, lymphocyte, esophagus, and soft tissue cancers, thereby limiting the scope of our analysis.”

3- *“Our method also identified KRAS-MAP3K2 as a potential pan-cancer synthetic lethality pair by comparing ERK pWT cell lines with KRAS mutant cell lines in the CRISPR screen.” - I don't understand this statement*

Given that the *KRAS* is associated with the ERK pathway, we would like to clarify that our reference to "ERK pWT cell lines" was intended to mean "*KRAS* pWT cell lines." To ensure terminological consistency throughout the manuscript, we have made the necessary modification and used *KRAS* pWT cell lines instead.

4- The authors detect SL pairs without and with 'leveraging pathway' data e.g. Figure 3C - it's not clear to me how the pathway data is used, either from the main text or the methods - this needs to be explained more clearly - initially, the paper describes 9 specific pathways where genes are altered but then focuses entirely on specific genes

We would like to clarify that our primary objective is to find SL interactions between genes not between pathways and genes. Specifically, we intend to find an interaction between a driver gene and an SL partner gene. We used the pathway data to categorized cell lines into two groups: wild-type and mutant cell lines, based on alterations in the driver gene and its associated pathways. In this classification, we leveraged pathway information to designate a cell line as wild-type not only if it lacked mutations in the driver gene but also in its associated pathways. This approach was implemented to minimize the potential influence of background genetic alterations in different cancer cell lines. We provided a comparison of detecting SL paris without and with leveraging pathway data in the manuscript.

5- Figure 3a - for each of the 9 pathways, useful to indicate how many cell lines have a mutation in any gene in that pathway, in addition to the existing view which shows each specific gene

Thanks for the suggestion. We have included Figure S3, which illustrates the distribution of cell lines exhibiting varying numbers of mutations in each signaling pathway.

6- Typos Abstract - 'MPA3K2' should be 'MAP3K2'

We have corrected this typo.

October 3, 2023

RE: Life Science Alliance Manuscript #LSA-2023-02268-TR

Dr. Hesam Montazeri
University of Tehran
Department of Bioinformatics
Ghods 37
Tehran, Islamic Republic of Iran

Dear Dr. Montazeri,

Thank you for submitting your revised manuscript entitled "Pathway-driven Analysis of Synthetic Lethal Interactions in Cancer Using Perturbation Screens". We would be happy to publish your paper in Life Science Alliance pending final revisions necessary to meet our formatting guidelines.

- please add ORCID ID for the secondary corresponding author--they should have received instructions on how to do so
- please add the Twitter handle of your host institute/organization as well as your own or/and one of the authors in our system
- please add an Author Contributions section to your main manuscript text
- please make sure the manuscript sections are aligned in accordance with LSA's formatting guidelines: please separate the Figure legends and Supplemental Figure legends into separate sections
- please upload your table S1 separately
- please add callouts for Figures 4E; 5D; S3A-B; S5A-B; S6B; S7A-C...S10A-C; S11A-B to your main manuscript text
- the information from the Supplemental Information file should be incorporated into the main manuscript text, along with those References into the main Reference list

A. FINAL FILES:

B. MANUSCRIPT ORGANIZATION AND FORMATTING:

Sincerely,

October 10, 2023

RE: Life Science Alliance Manuscript #LSA-2023-02268-TRR

Dr. Hesam Montazeri
University of Tehran
Department of Bioinformatics, IBB
Ghods 37
Tehran
Iran, Islamic Republic of

Dear Dr. Montazeri,

Thank you for submitting your Methods entitled "Pathway-driven Analysis of Synthetic Lethal Interactions in Cancer Using Perturbation Screens". It is a pleasure to let you know that your manuscript is now accepted for publication in Life Science Alliance. Congratulations on this interesting work.

DISTRIBUTION OF MATERIALS:

Again, congratulations on a very nice paper. I hope you found the review process to be constructive and are pleased with how the manuscript was handled editorially. We look forward to future exciting submissions from your lab.

Sincerely,
